# Chilling Requirement Validation and Physiological and Molecular Responses of the Bud Endodormancy Release in *Paeonia lactiflora* ‘Meiju’

**DOI:** 10.3390/ijms22168382

**Published:** 2021-08-04

**Authors:** Runlong Zhang, Xiaobin Wang, Xiaohua Shi, Lingmei Shao, Tong Xu, Yiping Xia, Danqing Li, Jiaping Zhang

**Affiliations:** 1Genomics and Genetic Engineering Laboratory of Ornamental Plants, Institute of Landscape Architecture, Department of Horticulture, College of Agriculture and Biotechnology, Zhejiang University, Hangzhou 310058, China; zhangrunlong@zju.edu.cn (R.Z.); xiaobinwang@zju.edu.cn (X.W.); 12116125@zju.edu.cn (L.S.); 22016252@zju.edu.cn (T.X.); ypxia@zju.edu.cn (Y.X.); 2Zhejiang Institute of Landscape Plants and Flowers, Hangzhou 311251, China; shxh2021@aliyun.com

**Keywords:** abscisic acid, chilling requirement (CR), endodormancy release, herbaceous peony, *Paeonia lactiflora* Pall, reactive oxygen species, underground bud dormancy

## Abstract

The introduction of herbaceous peony (*Paeonia lactiflora* Pall.) in low-latitude areas is of great significance to expand the landscape application of this world-famous ornamental. With the hazards of climate warming, warm winters occurs frequently, which makes many excellent northern herbaceous peony cultivars unable to meet their chilling requirements (CR) and leads to their poor growth and flowering in southern China. Exploring the endodormancy release mechanism of underground buds is crucial for improving low-CR cultivar screening and breeding. A systematic study was conducted on *P. lactiflora* ‘Meiju’, a screened cultivar with a typical low-CR trait introduced from northern China, at the morphological, physiological and molecular levels. The CR value of ‘Meiju’ was further verified as 677.5 CUs based on the UT model and morphological observation. As a kind of signal transducer, reactive oxygen species (ROS) released a signal to enter dormancy, which led to corresponding changes in carbohydrate and hormone metabolism in buds, thus promoting underground buds to acquire strong cold resistance and enter endodormancy. The expression of important genes related to ABA metabolism, such as *NCED3*, *PP2C*, *CBF4* and *ABF2*, reached peaks at the critical stage of endodormancy release (9 January) and then decreased rapidly; the expression of the *GA2ox8* gene related to GA synthesis increased significantly in the early stage of endodormancy release and decreased rapidly after the release of ecodormancy (23 January). Cytological observation showed that the period when the sugar and starch contents decreased and the ABA/GA ratio decreased was when ‘Meiju’ bud endodormancy was released. This study reveals the endodormancy regulation mechanism of ‘Meiju’ buds with the low-CR trait, which lays a theoretical foundation for breeding new herbaceous peony cultivars with the low-CR trait.

## 1. Introduction

Bud dormancy is an important survival strategy of temperate perennial plants in response to detrimental environments, such as cold in winter. With global climate change, bud dormancy has gradually become a popular topic in plant growth and development research fields in recent decades [1,2]. The complete process of bud dormancy includes three contiguous stages: paradormancy, endodormancy and ecodormancy [3]. Paradormancy is well accepted as the cessation of bud growth after shoot elongation stops. As the daylength shortens and the temperature decreases, perennials defoliate, and their buds generally transfer from paradormancy to endodormancy [4]. In autumn, with a decrease in temperature or day length, buds enter endodormancy, at which stage they are unable to sprout even under favorable conditions until the indispensable chilling accumulation has been completed [5,6]. Ecodormancy is the dormancy caused by adverse environmental factors such as low temperature and drought stress [3]. Once the environmental conditions are suitable, dormancy is released. However, endodormancy cannot be released before accomplishing indispensable chilling accumulation. Due to the global warming trend, many plants with winter dormancy traits suffer from hot autumns and warm winters, which disrupt normal endodormancy release because of insufficient chilling accumulation and then negatively affect their subsequent growth and development. Sufficient daylength and low temperature are the two most important environmental factors affecting the bud dormancy process. Short daylength could induce *Populus* L. and *Morus alba* L. to enter dormancy, and a low-temperature environment could induce *Pyrus* spp. and *Malus pumila* Mill. to enter dormancy [7,8,9]. Shortening daylength alone does not induce these fruit trees to enter bud dormancy without decreasing temperature. The dormancy of *Vitis vinifera* L. and *Armeniaca vulgaris* Lam. was mainly controlled by sunshine duration and low temperature [10,11].

The internal factors controlling dormancy processes are complicated and involve carbohydrate conversion, hormone signaling, antioxidant metabolism, water transport, gene regulation, and DNA methylation and demethylation [12,13,14]. During the paradormancy stage, sugar metabolic activity is continuously vigorous, and the large amount of accumulated soluble total sugar and sucrose not only helps to improve cell osmotic potential to resist the cold in winter but also acts as a regulatory substance to inhibit the growth of buds in perennials [15,16,17]. In addition, the content and balance of endogenous hormones are currently widely accepted to control the occurrence and release of bud endodormancy. A previous study showed that the formation and maintenance of abscisic acid (ABA) and the status transition of bud dormancy were closely interrelated in potato. The ABA content increased with the deepening of bud endodormancy but decreased regularly with the endodormancy release [18]. Generally, the ABA content continuously increased in paradormancy, reached the highest level in endodormancy and finally decreased to the lowest level in ecodormancy [18,19,20]. Gibberellin (GA) was confirmed to promote the dissolution of dormancy. The accumulation of low temperature in winter resulted in an increase in GA content and then promoted the degradation of callose in the sieve tube, which was conducive to promoting the release of endodormancy [21]. Additionally, reactive oxygen species (ROS) function as signal transduction factors and are also involved in many regulatory processes related to plant growth and development. The activities of superoxide dismutase (SOD) and peroxidase (POD) gradually increased, while catalase (CAT) activity showed the opposite trend after the end of endodormancy in pear [22].

In addition to the control of physiological factors, bud endodormancy induction, transition and release are also regulated by various genes and associated pathways. Hormone signaling and metabolism-related genes, such as 9-*cis*-epoxycarotenoid dioxygenase (*NCED*), ABA-insensitive 5 (*ABI5*), protein phosphatase 2ca (*PP2C*), and *Arabidopsis thaliana* gibberellin 2-oxidase (*GA2**ox*)*,* are popular research topics in plant bud endodormancy, while sucrose synthase 3 (*SUS3*), *Arabidopsis thaliana* cell wall invertase 1 (*CWINV*), cyclin D3 (*CYCD*) and cyclin-dependent kinase b2;2 (C*DKB22*) are also frequently involved in carbohydrate and ROS metabolism. These genes promote or inhibit plant bud endodormancy and may play an important role in breeding plant germplasm with long or short endodormancy periods through gene overexpression and silencing. In addition, *dormancy-associated MADS-box* (*DAM*) genes belong to the *MADS-box* gene family and are crucial for bud endodormancy regulation [23]. *DAM* genes were isolated and identified in a study on bud endodormancy in peach with rare evergreen traits [24]. Since then, researchers have identified related genes in many other species (pear, plum, kiwifruit) and further confirmed their association with the endodormancy process [25,26,27]. These genes form a complex dormancy regulatory network with other *MADS-box* genes and genes related to carbohydrate, phytohormone and antioxidant metabolism [28].

The bud endodormancy of *Paeonia suffruticosa* belonging to the same genus as *Paeonia lactiflora* has been studied for decades. During the process of herbaceous peony flower bud endodormancy release, accompanied by the hydrolysis of soluble sugar, starch, and the increase of antioxidant enzyme activity [29]. Hormone regulation of herbaceous peony flower bud endodormancy is mainly dependent on the ratio of endodormancy release promoter to inhibitor, especially GA and ABA, the ratio between the two is closely related to the endodormancy process of herbaceous peony flower bud [18,30,31]. In addition, in-depth research at the molecular level has mainly focused on methylation and gene regulation [32,33]. GA and low temperature both have the effect of reducing DNA methylation. The gene *GA2ox8* related to GA synthesis is also one of the important genes regulating endodormancy [34].

Herbaceous peony (*Paeonia lactiflora* Pall.) is a world-famous ornamental plant that originates from China and has a long cultivation history worldwide [35]. It is widely used in cut flowers, potted plants and gardens. This species is mainly distributed in temperate and cool regions of the Northern Hemisphere, such as Europe, North America and Asia [36]. To survive in cold winters, the aboveground parts of herbaceous peony wither in autumn, and the underground bud undergoes endodormancy. Only after accumulating enough of a chilling requirement (CR) in winter can it grow well in the coming year. Therefore, chilling is an essential factor in the endodormancy-growth cycle of herbaceous peony [37]. In this dormant period, the underground buds of herbaceous peony undergo a series of physiological changes, such as carbohydrate conversion, hormone metabolism and ROS signal transduction [38]. The introduction of herbaceous peony from high latitudes to low latitudes is beneficial to expand its distribution range and increase its application in landscaping. However, the winter temperature at low latitudes is too high to meet the CR, which poses a great obstacle to the release of bud endodormancy in the coming spring [39,40].

In this study, based on previous research, we selected the low chilling requirement herbaceous peony cultivar ‘Meiju’ as the research material and carried out further evaluation and confirmation of the chilling requirement and observation of growth morphology and bud endodormancy (including carbohydrate metabolism, hormone metabolism, ROS signal transduction and other physiological indices and related gene expression). Based on the morphological index, physiological data and molecular experimental results, a possible regulatory network of endodormancy release in underground buds of ‘Meiju’ was studied, which laid some theoretical foundations for breeding cultivars with low cold demand and introducing *Paeonia lactiflora* from high latitudes to low latitudes. Our experiment could enrich the studies on bud endodormancy of herbaceous peony in low latitude regions.

## 2. Results

### 2.1. Morphology and CR Evaluation during Bud Dormancy and Sprouting

#### 2.1.1. Morphological Observations of Bud Dormancy and Sprouting

As the natural temperatures decreased, the aboveground parts of herbaceous peony ‘Meiju’ withered gradually, In autumn, initial periods of more than 50% leaves losing green/withering from 2 to 14 August, and the underground buds of ‘Meiju’ entered the paradormancy phase until a sufficient chilling accumulation the next spring. The new buds (Figure 1) sprouted around 7–13 March in outdoors, and the newly unearthed buds were dark red. With the strengthening of photosynthesis, the buds gradually turned into green leaves.

As shown in Figure 2A,B, the temperature in Hangzhou began to decrease gradually in November 2018, reached the lowest value in January, and gradually increased in late February. Under this temperature change, the aboveground parts of the herbaceous peony began to wither in September and the underground buds went dormant; the color became bright red in March of the following year, and then buds began to germinate. Figure 3 shows that with the gradual decrease in temperature, the rhizome of *Paeonia lactiflora*, which was moved into the greenhouse in batches, showed a gradually increasing trend for BPF, ANS, APW, APH, and ADS, while WFS and WAS gradually decreased with subsequent germination and growth. The six indicators all tended to be stable on Jan. 9, and the subsequent changes were not significant overall.

#### 2.1.2. Comprehensive Evaluation of the CR of ‘Meiju’ to Break Bud Endodormancy

In the cytological observation of the endodormancy transition and release period of underground buds, it was found that during paradormancy, the arrangement of cells in underground buds was loose and the division activity was not significant, but in the endodormancy release stage (9 January), the cells were arranged closely, the nuclear color deepened, the cell division activity was high, and the cell states in leaf primordia, growing points, vascular strand primordia and dud primordia were obviously different from those in the previous period (Figure 4). By 26 February, the cells had elongated, and the morphological structure had undergone great changes compared with the endodormancy stage (14 November). Obvious vascular bundle cells have been produced, and different cell morphologies appeared (Figure 4).

In addition, the date when the daily accumulation of chill units (DACU) (Figure 2B) increased and first became a positive value (8 November 2018) was chosen as the start date for evaluation by the UT model (Figure 2B). Due to the limited number of slices, the slice acquired on 14 November was adopted to show the bud cell status near the starting point of the UT model evaluation. During this period, the differentiation of ‘Meiju’ flower buds was not completed (Figure 2C). The dates when the potted plants were moved to the greenhouse are shown in the abscissae in Figure 3. With the delay in the moving date, chilling accumulated gradually, and most of the morphological indices relevant to bud dormancy, sprouting and growth increased and tended to be stable on 9 January, which shows that the accumulation of CR (667.5 CUs) in ‘Meiju’ has been completed on 9 January during the winter from 2018 to 2019. After that, ‘Meiju’ entered the ecodormancy stage (Figure 2).

For the UT model, the start times were defined as 0:00 am on the day when the daily chill-unit accumulation increased and first became positive after September (8 November) [37]. In term of the morphological changes, it was confirmed that 9 January was the endodormancy elimination period. According to the definition of the end point of the chilling accumulation assessment, 9 January was also the end point of the cold capacity assessment. Using the UT model, the cumulative value from the starting point to the end point was 677.5 CUs.

### 2.2. Physiological Responses of the Underground Buds during the Whole Dormancy Process

#### 2.2.1. Change in the Carbohydrate Contents

The contents of sucrose, total soluble sugar and starch were measureded to study the carbohydrate metabolism of the dormant ‘Meiju’ buds in this study. Sucrose and total soluble sugar contents increased gradually, reached a peak on 9 January when the CR was satisfied, and then decreased slowly. The starch content decreased slowly during the whole dormancy period (Figure 5).

#### 2.2.2. Change in the Phytohormone Contents

The contents of trans-Zeatin-riboside (ZR), indole-3-acetic acid (IAA), GA_3_ and brassinolide (BR) all first declined and then rose overall, and their trends fluctuated to different degrees. Among them, ZR, IAA and GA_3_ (nearly) reached their lowest values on 9 January when the CR of ‘Meiju’ was satisfied for release bud dormancy. However, the contents of ABA and methyl jasmonate (JAME) increased gradually in the early stage and then decreased rapidly and significantly after reaching the highest level on 26 December (Figure 6). There was a similar trend between the two hormones, with both reaching a peak at the critical stage of release dormancy.

#### 2.2.3. Change in the Enzyme Activity of Antioxidants

To understand ROS metabolism in the process of bud dormancy and release in ‘Meiju’, the activities of the antioxidant enzymes SOD, POD and CAT and the contents of malondialdehyde (MDA) were observed. Overall, SOD activity first decreased and then increased, while POD activity showed the opposite trend; CAT activity increased sharply during endodormancy and was finally maintained at the level of 40–50 ng/g FW; and the MDA content increased slightly (Figure 7). Taking a comprehensive view of the four indices, the activities of SOD, POD and CAT all reached their extreme values on 9 January [41].

### 2.3. Correlation Analysis among All Morphological and Physiological Indices

Indices related to carbohydrate metabolism (Sucrose, Total Soluble Sugar and Starch), plant hormones (ZR, GA_3_) and the ROS system (SOD, CAT) have high correlation coefficients and could be used as important indicators for follow-up research (Table 1).

### 2.4. Expression Analysis of Representative Dormancy-Related Genes in ‘Meiju’ Underground Buds

#### 2.4.1. Selection and Annotation of the Representative Dormancy-Related Genes

Since no high-quality reference genome of the species *Paeonia lactiflora* Pall. exists, the Arabidopsis genome was adopted to annotate the homologous unigenes from underground buds during endodormancy to ecodormancy based on our previous transcriptomic sequencing of ‘Meiju’(unpublished) (Table 2).

#### 2.4.2. Expression of Genes Related to Carbohydrate Metabolism in Different Dormancy Stages

Genes related to carbohydrate transport and metabolism (α-amylase-like 3 (*AMY3*), cell wall invertase 1 (*CWINV1*), starch synthase 3 (*SS3*)) and genes related to cell development (sucrose phosphate synthase 1f (*SPS1F*), sucrose synthase 3 (*SUS3*)), their expression levels were up-regulated gradually and reached the highest level on 23 January, then decreased rapidly. However, the expression of phosphofructokinase 3 (*PFK3*) decreased gradually before 23 January. β-Amylase 3 (*BMY3*) and APL3 genes were not significantly differentially expressed before 9 January, and remained at a low expression level. Their expression levels were significantly upregated on 9 January, and then decreased rapidly. Both β-glucosidase (*BGLU*) and UDP-glucose pyrophosphorylase 2 (*UGP2*) were expressed the lowest on 9 January (Figure 8).

#### 2.4.3. Expression of Genes Related to Antioxidant Metabolism in Different Dormancy Stages

The expression levels of *Arabidopsis thaliana* texpansin 6 (*EXPA6*), *CDKB22*, cyclin d3 (*CYCD3*) and histone H1-3 (*HIS1-3*) decreased to the lowest level on 9 January and then gradually increased. In addition, the expression levels of expansin-like a2 (*EXLA2*), *Arabidopsis thaliana* monothiol glutaredoxin 17 (*GRXS17*), β-1,3-glucanase 3 (*BG3*), ARP6 and catalase 2 (*CAT2*) were briefly down-regulated and then gradually increased, and were at a high level on 9 January or 23 January. In the early stage of endodormancy, the expression level of peroxidase 52 (*PER52*) decreased significantly, and did not rise after reaching the lowest level on 9 January (Figure 9).

#### 2.4.4. Expression of Genes Related to Hormone Metabolism in Different Dormancy Stages

The expression levels of 9-*cis*-epoxycarotenoid dioxygenase 3 (*NCED3*) and 9-*cis*-epoxycarotenoid dioxygenase 3 (*NCED4*) related to ABA synthesis were always low before endodormancy release, but significantly increased after 9 January. The reason for this phenomenon may be that different genes in the large gene family of *NCEDs* play different roles in ABA synthesis [42]. In addition, C-repeat-binding factor 4 (*CBF4*), abscisic acid responsive elements-binding factor 2 (*ABF2*) and *ABI5* genes involved in ABA signal transduction and metabolism also had different expression levels at different stages. *CBF4* and ABF2 were highly expressed when the endodormancy was released, while *ABI5* was less expressed at this time (Figure 10). The expression levels of genes involved in gibberellin synthesis and metabolism (such as GAST1 protein homolog 4 (*GASA4*) and *Arabidopsis thaliana* gibberellin 2-oxidase 8 (*GA2ox8*)) were significantly increased after the endodormancy was released. The expression level of repressor of GAL-3 1 (*RGA1*) gradually decreased before 9 January.

#### 2.4.5. Changes in the DAM-SOC1-AP1 Pathway and Others in Different Dormancy Stages

Figure 11 shows the expression patterns of other genes that also probably participate in the bud dormancy transition and release of herbaceous peony. The expression levels of ga insensitive dwarf1a (*GID1A*) and phytochrome interacting factor 3 (*PIF3*) reached their highest level on 12 December when the buds were still in the deep phase of endodormancy. The expression of *CBF4* was significantly increased to its highest level on 9 January, *SOC1* is another name for agamous-like 20 (*AGL20*), which belongs to the *MIKC-type MADS-box* gene family with short vegetative phase *SVP* and is considered to have the function of promoting the endodormancy release and vegetative growth of buds. Their expression increased significantly to the highest point between 9 January and 23 January (Figure 11) [43]. The expression levels of *SPT*, *RGA1* and *SPY* decreased significantly before 9 January.

## 3. Discussion

### 3.1. CR Evaluation Is an Important Method to Popularize Herbaceous Peony at Low Latitudes

Compared with long-term breeding, CR evaluation and screening of cultivars with low cold demand is more direct and can help select target cultivars in a shorter time. Typical materials with high cold demand and low cold demand can be selected after evaluation and can be used as parents for scientific research and long-term breeding [44]. For herbaceous peonies, it will take a long time to create new cultivars with the low-CR trait by a traditional crossbreeding strategy. Therefore, the introduction of herbaceous peonies from high-latitude areas and CR evaluation are needed to screen for low-CR cultivars [37]. In this study, the CR of ‘Meiju’ was calculated as 677.5 CUs by the UT model, and thus ‘Meiju’ was verified as a low-CR cultivar compared with other northern cultivars (‘Zhuguang’: 1182 CUs, ‘Qiaoling’: 1017 CUs, ‘Qihua Lushuang’: 685.5 CUs, ‘Fen Yunu’: 685.5 CUs [37]). In our previous study, the bud endodormancy of ‘Meiju’ was released on 9 January 2017 and the CR was correspondingly calculated as 1017.0 CUs based on the UT model [37]. As a matter of fact, we analyzed the morphological data more carefully afterwards, and deduced that the bud endodormancy of ‘Meiju’ had probably been released on 26 December 2016. Therefore, the true CR value of ‘Meiju’ should be 685.5 CUs during the winter of 2016 to 2017, which is very close to 677.5 CUs verified in this study. This northern cultivar grew vigorously and bloomed exuberantly in Hangzhou based on multiyear observations; thus, it could be used as a pioneer cultivar to popularize herbaceous peony in Hangzhou and other low-latitude areas [37,41].

### 3.2. Carbohydrate Metabolism Participates in the Regulation of Endodormancy Release

The conversion of sucrose, total soluble sugar and starch is an important factor in regulating bud endodormancy in perennials [15,16,17,45,46]. Generally, in the stage of endodormancy, starch tends to decrease, while sucrose and total soluble sugar increase accordingly. Starch is a polysaccharide that cannot be directly used by plants, and it can only be used when it is converted into monosaccharides. Therefore, the increase in sucrose and total soluble sugar content in herbaceous peony buds before 9 January may come from the decomposition of starch, which can provide energy for the life activities of herbaceous peony buds and promote the gradual release of endodormancy in herbaceous peony buds [15,16]. However, this is different from the results from a study in spinach, which shows that different species have different regulatory mechanisms [47].

On the other hand, the increase in sucrose and total soluble sugar content may also help to reduce the osmotic pressure of cells and ensure that the cells can fully absorb water, thus increasing the cold resistance of herbaceous peony buds and helping them survive the lowest temperature period in Hangzhou in January (Figure 2) [48].

### 3.3. Relationship between Hormone Metabolism and Endodormancy Release

Plant endogenous hormones play an important role in the induction, transformation and release of bud endodormancy [4,49,50]. Plant endogenous hormones mainly include two categories: promoting the endodormancy release of buds (such as GA, ZR, BR, JA, and IAA) and inhibiting the release (maintaining endodormancy, such as ABA) [12,51,52]. The endogenous hormone content in plants is an important factor regulating the induction and release of endodormancy, and the dynamic balance between hormone contents is often highly correlated with the process of dormancy [4,50].

Figure 10 shows that among the genes associated with ABA, only the expression pattern of *ABI5* was completely consistent with the change in ABA level from endodormancy to ecological dormancy. *ABI5* is a key transcription factor in the ABA signaling pathway that helps to promote ABA signal transduction [44]. The rapid downregulation of *ABI5* around 9 January may be one of the reasons for the decrease in ABA content, which is the key hormone-related gene to accelerate the release of endodormancy [53].

As an ABA binding factor, *ABF2* often combines with *AREB1* to form the *AREB1/ABF2* transcription factor, which is the key factor downstream of the ABA signaling pathway [54,55]. In the process of bud dormancy, the expression of *ABF2* is theoretically conducive to ABA signal transduction, thus maintaining bud endodormancy [6,56]. In this study, *ABF2* showed a unimodal curve, which reached its peak on 9 January (Figure 10), suggesting that its high expression promoted the release of endodormancy in buds. However, the trend of *ABF2* expression during the whole endodormancy period was opposite that of ABA (Figure 6 and Figure 10), suggesting that its expression was upregulated but that the ABA content was decreased.

However, the expression of other genes is only in line with expectations at some stages. For example, the expression of upstream genes *NCED3* and *NCED4* in the ABA synthesis pathway increases during endodormancy (Figure 10, 17 October–9 January); this gene has been proven to be an important gene for promoting ABA synthesis, which is consistent with the results of ABA content determination (Figure 6) [31,57,58]. One of the possible reasons why ‘Meiju’ is a low CR herbaceous peony is that the *NCED* gene is upregulated in early winter. However, in the stage of ecodormancy in which the content of ABA decreased, *NCED3* and *NCED4* continued to be upregulated and maintained at a relatively high expression level (9 January–27 February). Similar results are also reflected in pear, which shows that the genes in the *NCED* gene family regulate different biological processes, and these genes may also be involved in the resistance of herbaceous peony buds to low temperature in winter during ecodormancy [4].

GA3 is one of the important hormones regulating bud endodormancy [59]. In this study, the content of GA3 was gradually decreased before the endodormancy release, and the decrease might be related to the antagonism of ABA. In the stage when endodormancy release (before 9 January), the ABA content decreased rapidly and the ratio of GA/ABA began to increase, which promoted the release of endodormancy. In terms of gene expression, *GA2ox* has been proved to have a negative regulatory effect on GA synthesis in *Arabidopsis thaliana* and poplar [60,61]. After the endodormancy released, the rapid increase in GA3 content might be related to the up-regulation of *GASA4* gene expression, which is related to gibberellin synthesis.

In summary, the chilling requirement of ‘Meiju’ is shorter, and the endodormancy release period is earlier, which is closely related to the early decrease in ABA content and the regulation of related genes such as *ABI5*, *NCED3*, and *PP2C* [42,58,62,63]. More gene function verification experiments are needed to confirm this hypothesis. In addition to ABA, hormones that promote the release of bud endodormancy, such as GA_3_, IAA, JA, ZR and BR, which act in opposition to ABA, can also be involved in determining the characteristics of low chilling requirements [64,65,66,67].

### 3.4. Antioxidant Metabolism Involved in Endodormancy Release Regulation

Many studies have shown that antioxidant metabolism is also involved in the regulation of bud endodormancy [22,68]. Similar to a study in pears, the activity of SOD decreased before the release of ecodormancy but began to increase after this release, and the activity of CAT gradually increased before the termination of endodormancy and began to decline after termination [22]. The activity of POD was upregulated rapidly before the release of endodormancy and then decreased rapidly. These findings are completely different from those in pears [22]. The enzyme activity related to the antioxidant metabolic pathway responded rapidly during paradormancy and coincided with the degree of dormancy. This suggests that ROS may act as a signal transduction substance to sense environmental changes and participate in the regulation of paradormancy induction and release. However, the related regulatory mechanisms may be different in different species [22,69].

At the level of gene expression, the expression of most genes related to antioxidant metabolism (*EXPA6*, *CDKB22*, *CYCD3*, *BG3*, actin-related protein 6 (*ARP6*), *PER52*, histone h1-3 (*HIS1-3*)) decreased gradually before the release of endodormancy and then increased again (Figure 9). The expression of these genes changed dramatically, decreased rapidly when entering endodormancy, and then increased rapidly after endodormancy was released.

In other metabolic pathways, the expression trend of genes related to ROS signal transduction was opposite those related to sucrose content (Figure 5), total soluble sugar content (Figure 5) and ABA content (Figure 6). ROS may play a role in sensing and transmitting signals in the process of dormancy release in buds of ‘Meiju’. This shows that the efficient signal transmission of ROS and the timely reflection of other physiological activities may be one of the reasons for the low cooling requirement of ‘Meiju’.

### 3.5. Judgment of Endodormancy Release Should Be Based on Comprehensive Research

According to the experiment of moving outdoor potted herbaceous peony to a greenhouse, several key indices (BPF, ANS, APW, APH, ADS, WFS) were used as the most intuitive evidence to judge that the endodormancy of herbaceous peony was released around 9 January.

Additionally, the state of cells in leaf primordia (LP), growing point (GP), vascular strand primordia (VS), bud primordia (BP) and other parts is also quite different from that in the previous period, with an obvious tight arrangement of cells and a deeper color of nuclei (Figure 4). The continuous accumulation of these soluble carbohydrates possibly helped to increase the osmotic potential of underground bud cells and enhanced the cold resistance of ‘Meiju’ (Figure 5). High concentrations of sugar, together with ROS signaling, may lead to a decrease in the contents of hormones such as ABA, JAME, and IAA (Figure 12). The expression of many important genes related to endodormancy release, such as *CBF4*, *PP2C*, and *NCEDs*, reached the highest level around 9 January (Figure 10) [41,47,48,50].

These indices may be used as potential judgment indices of CR satisfaction and the endodormancy release period to provide some new judgment indices for the general method of endodormancy release when the germination rate reaches 50%, but they need to be further studied and verified [70].

### 3.6. Possible Regulatory Networks for Dormancy Release of Underground Buds in Herbaceous Peony

Temperature is closely related to the induction, maintenance and release of perennial bud endodormancy [45,71]. It has been proven that there is a relationship between cold signal transduction and molecular networks related to bud endodormancy in some temperature-sensitive plants [72]. There is a great similarity between these plants and ‘Meiju’. In the underground bud of ‘Meiju’, the change in external temperature causes ‘Meiju’ to produce a large amount of MDA, causing damage to the bud of “Meiju”.ROS signal transduction responded quickly and transmitted signals to other related metabolic processes in time, among which glucose metabolism and hormone metabolism are the most important. The ABA content increased rapidly through the regulation of the *NCED3* and *NCED4* genes. In the later stage of endodormancy, *GA2ox8* and other genes regulated gibberellin accumulation and inhibited the effect of ABA. Under the joint regulation of the DAM gene, the endodormancy of ‘Meiju’ underground buds was released (Figure 12).

A large number of antioxidant metabolism-related genes and some carbohydrate metabolism-related genes (*SS3*, *BGLU*) responded quickly (Figure 12). Which made the sugar and hormone metabolism extremely active, allowing earlier completion of the accumulation of soluble sugars and the synthesis and metabolism of endodormancy release inhibitor (ABA). Therefore, the ABA/GA ratio decreased rapidly in the earlier period (9 January), which may be an important reason for the short endodormancy period and low cooling capacity of ‘Meiju’.

## 4. Materials and Methods

### 4.1. Plant Materials

Based on previous research, *Paeonia lactiflora* Pall. ‘Meiju’ was selected as the research material to perform all experiments in this study [37,41]. ‘Meiju’ was introduced from Heze City (E 34°39′–35°52′, N 114°45′–116°25′), Shandong Province, China. The height of ‘Meiju’ was about 60–80 cm, its sprouting rate could exceed 50% outside in March, and the leaves unfold 1–2 weeks after sprouting. Usually in early August, the leaves wither [41]. This cultivar has purple-red buds, tender shoots, leaves and petals, an anemone flower type, thick fragrance (Figure 1 and Figure 13), and relatively low CR traits (Compared with many northern herbaceous peony cultivars, ‘Meiju’ has lower chilling requirement in winter and shorter endodormancy period. Its buds Sprouted earlier.) [70]. Thus, ‘Meiju’ has high ornamental value and good potential for growth in low-latitude areas such as Hangzhou. ‘Meiju’ was planted outside at the Perennial Flower Resources Garden of Zhejiang University in Hangzhou (E 118°21′–120°30′, N 29°11′–30°33′). No artificial intervention was carried out at any stage of growth and development, and natural illumination was used.

### 4.2. Determination of the Dormancy Release Date and Morphological Observations

Potted ‘Meiju’ crowns with enough plump buds were placed in a nursery under natural low temperature and then moved to a greenhouse (15–25 °C, 80% relative humidity, regular watering and fertilizer applications) at intervals of two to four weeks. The specific transfer dates were 14 November 2018, 12 December 2018, 26 December 2018, 9 January 2019, 23 January 2019, 13 February 2019, and 27 February 2019. Regrowth performance was observed regularly every week. The main observation indices are as follows: the number of weeks until the first plant sprouted in the glasshouse (WFS), the number of weeks until all plants sprouted in the glasshouse (WAS), the bud break percentage five weeks after the potted crowns were transported to the glasshouse (BPF), the average number of mature and normal stems (ANS) and the average diameter of mature and normal stems (ADS), average plant width during the full-flowering period (APW), average plant height during the full-flowering period (APH). All were used to determine the period of release of ‘Meiju’ endodormancy.

### 4.3. Preparation of Paraffin Sections

Paraffin slices were dewaxed in water: the slices were successively put into xylene I for 20 min, xylene II for 20 min, anhydrous ethanol I for 5 min, anhydrous ethanol II for 5 min, and 75% alcohol for 5 min, followed by washing with tap water.

Sections were stained with safranin-fixed green dye and then dehydrated with anhydrous ethanol. Finally, the slices were dehydrated and sealed with anhydrous ethanol I for 5 min, anhydrous ethanol II for 5 min, anhydrous ethanol III for 5 min, xylene for 5 min, and xylene II for 5 min with transparent and neutral gum. Finally, microscopy examination, image acquisition and analysis were performed.

### 4.4. Selection of an Evaluation Model

With regard to the selection of the CR evaluation model of *Paeonia lactiflora*, we selected the UT model as the final CR evaluation model among the UT model, NC model and LC model. The start times were defined as 0:00 am on the day when the daily chill-unit accumulation increased and first became positive after September (Figure 2B). The CR evaluation ended when the potted crowns were moved into the glasshouse. The CR results represented the cumulative values of effective chill hours or units between the start and end times evaluated by UT models [37]. For more information about the starting point of the cooling capacity assessment, please refer to Wang et al. [37]. Although the UT model is not considered suitable for the CR assessment of some tropical fruit trees, it seems to be suitable for herbaceous peony assessment in Hangzhou and other low latitude areas, mainly because of the reasonable start time of assessment, stable CR value and high correlation with most morphological indices.

### 4.5. Determination of Related Physiological Indices and Gene Expression

For the determination of physiological indices related to ‘Meiju’, we selected some data that are very related to the dormancy process of underground buds, such as sugar content, hormone content and antioxidant enzyme activity, and analyzed their contents in different growth stages to reflect the dormancy of ‘Meiju’ underground buds. Then, through transcriptome analysis, combined with the annotation of homologous genes in *Arabidopsis thaliana*, as well as previous research progress, we screened some important genes for expression detection.

#### 4.5.1. Measurements of the Soluble Sugar Contents

The contents of soluble sugar and starch were determined by anthrone-sulfuric acid colorimetry. A 0.3 g ‘Meiju’ underground bud sample was fully ground with a small amount of quartz sand and 5 mL of 80% ethanol. Then, the homogenate was transferred to a 10 mL centrifuge tube and centrifuged at 4 °C and 4000 rpm centrifugation for 5 min. This process was repeated three times, and after each step, the supernatant was removed and 5 mL of ethanol was added. Finally, the supernatant was collected in a 50 mL volumetric flask and the volume was adjusted to 50 mL for the determination of sugar content.

Next, 0.2 mL of the sample to be tested was collected, and 0.8 mL of H_2_O and 0.2 mL of 30% KOH were added to the tubes, which were placed in a water bath at 95 °C for 10 min. After that, 5 mL of anthrone was added, and the tubes were placed in a water bath at 90 °C for 15 min. The sucrose content was calculated according to the standard curve. Then, a 0.5 mL sample was collected, and 0.5 mL of H_2_O and 10 mL of anthrone were added; tube were placed in a boiling water bath for 10 min. The value at 620 nm OD was measured after cooling in the dark. The total soluble sugar content was calculated according to the standard curve.

#### 4.5.2. Measurements of the Starch Contents

The precipitate after centrifugation was added to 52% perchloric acid (5 mL), and the starch was extracted. The starch was centrifuged twice at 4 °C and 10,000 rpm for 30 min. After each centrifugation, the supernatant was transferred to a 50 mL capacity bottle, and deionized water was added to 50 mL to obtain a total of 0.1 mL sample, 0.9 mL H_2_O and 10 mL anthrone. Samples were placed in a boiling water shaker at 10 rpm and then moved to a dark ice bath and cooled rapidly to room temperature for 10 min for measurement at 620 nm. The starch content was calculated according to the standard curve.

#### 4.5.3. Measurement of Hormone Contents

The contents of underground bud hormones (JA, IAA, ABA, GA, BR and ZR) in ‘Meiju’ were determined by enzyme-linked immunosorbent assay (ELISA) at the School of Agronomy and Biotechnology, China Agricultural University. Samples of buds (three biological replicates) were ground in a mortar at 0 °C in 10 mL of 80% (*v/v*) methanol extraction medium containing 1 mM butylated hydroxy-toluene as an antioxidant. The extract was incubated at 4 °C for 4 h and centrifuged at 4800× *g* for 15 min at 4 °C. The supernatants were sequentially passed through Chromosep C18 columns and prewashed with 10 mL of 100% and 5 mL of 80% methanol. For detailed methods, please refer to Zhao et al. [73].

#### 4.5.4. Determination of ROS-Related Physiological Indices

Each index was determined by an assay kit as follows. The malondialdehyde (MDA) contents were measured with an MDA assay kit (A003-3, Jiancheng, China). A test method based on thiobarbituric acid was used. The SOD enzymatic activity was measured by an assay kit (A001, Jiancheng, China). The POD enzymatic activity was measured by an assay kit (A084-3, Jiancheng, China). The CAT enzymatic activity was measured by an assay kit (A007, Jiancheng, China). Each assay was replicated at least three times per sample.

#### 4.5.5. Total RNA Extraction and Quantitative Real-Time PCR (qRT-PCR) Analysis

According to the instructions, total RNA was extracted from the underground buds of ‘Meiju’ with the RNAprep Pure Kit (Beijing, China). First strand cDNA was synthesized by a FastQuant RT Kit (TIANGEN, Beijing, China) and used as the template for the amplification experiment. The qRT-PCR polymerase chain reaction system was performed with specific primers. The final reaction volume was 10 μL (F-Primer 0.5 μL, R-Primer 0.5 μL, 20-fold dilution cDNA 4 μL). The thermal cycle conditions were as follows: 2 min at 95 °C and 39 cycles of 5 s at 95 °C and 30 s at 55 °C; 5 s at 95 °C, 5 s at 65 °C and 5 s at 95 °C.

### 4.6. Statistical Analysis

The experiment was organized into a completely random design. The data were subjected to ANOVA followed by least-significant difference tests and then tested for the lowest significant difference. The difference was statistically significant at *p* < 0.05 or *p* < 0.01. All the data were analyzed by using the social science statistics software package (SPSS v.26.0, IBM, Armonk, NY, USA). GraphPad Prism 8.0 (GraphPad Software, Inc., La Jolla, CA, USA) was used for figure construction.

## 5. Conclusions

In this study, *Paeonia lactiflora* was introduced from high latitudes to low latitudes to promote the open field application of herbaceous peony in gardens, cultivate herbaceous peony cultivars with low CR and overcome the obstacle of bud endodormancy.

In summary, the dynamic balance among sugar metabolism, ROS signal transduction and hormones is involved in the regulation of underground bud endodormancy, and we propose a ‘Meiju’ underground bud endodormancy regulation model (Figure 12). ABA and GA are the most important regulatory hormones, but the antagonistic mechanism between them and the complete mechanism of ‘Meiju’ underground bud endodormancy regulation need to be further studied.

## Figures and Tables

**Figure 1 ijms-22-08382-f001:**
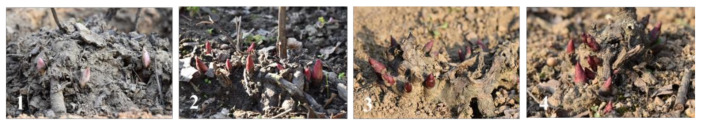
Growth of ‘Meiju’ buds at different stages. ‘Meiju’ was planted outside at the Perennial Flower Resources Garden of Zhejiang University in Hangzhou (E 118°21′–120°30′, N 29°11′–30°33′). (**1**): 10 November; (**2**): 15 December; (**3**): 11 January; (**4**): 6 February.

**Figure 2 ijms-22-08382-f002:**
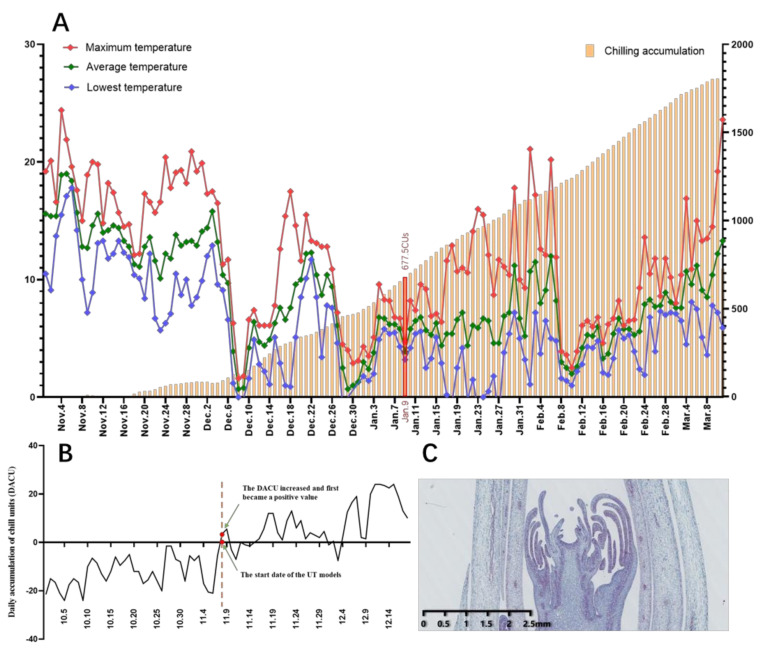
(**A**): Temperature change and chilling accumulation of underground buds from ‘Meiju’ during dormancy (using the UT model to calculate the chilling accumulation). The line chart represents the daytime temperature, and the bar chart represents the chilling accumulation value. (**B**): Trends in the daily chill-unit accumulation and start dates for CR evaluations via the UT model. (**C**): A slice of the underground bud closest to the start date.

**Figure 3 ijms-22-08382-f003:**
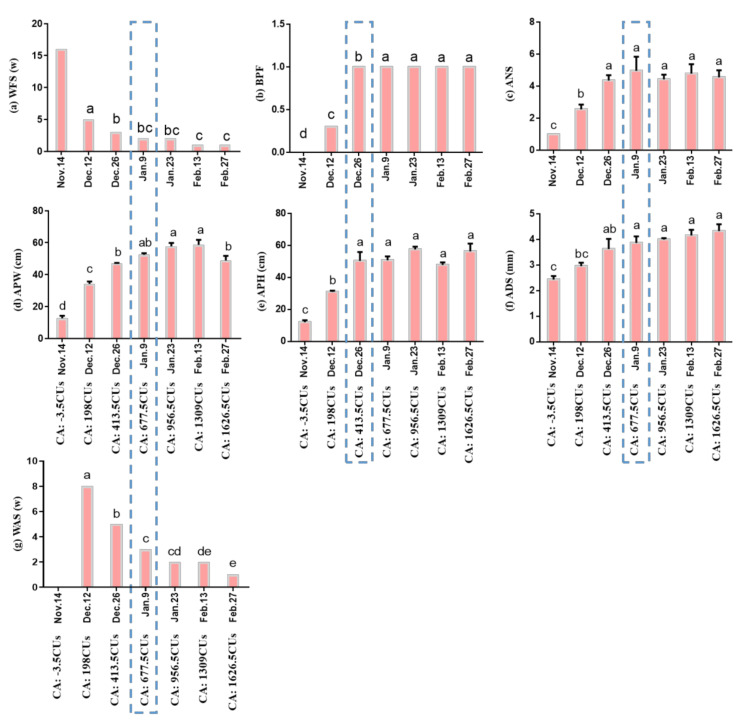
Sprouting and regrowth of the ‘Meiju’ plants transferred to the greenhouse on different dates (with different natural chilling accumulations). The blue dashed frames indicate the dates at which the change in morphological indices became sufficiently stable. (a) WFS: Number of weeks until the first plant sprouted in the glasshouse; (b) BPF: Bud break percentage five weeks after the potted crowns were moved to the glasshouse; (c) ANS: Average number of mature and normal stems during the full-flowering period; (d) APW: Average plant width during the full-flowering period; (e) APH: Average plant height during the full-flowering period; (f) ADS: Average diameter of stems during the full-flowering period; (g) WAS: the number of weeks until all plants sprouted in the glasshouse. Different letters indicate a significant difference among different dates (*p* < 0.05).

**Figure 4 ijms-22-08382-f004:**
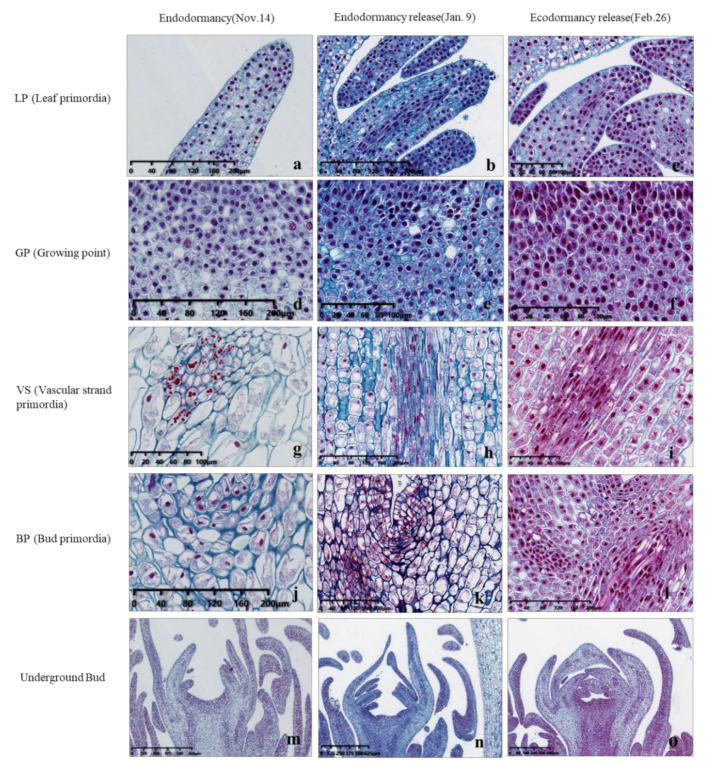
Cytological observation of ‘Meiju’ underground buds from endodormancy to ecodormancy. The dyeing method in the picture is as follows: Safranin O-Fast Green staining. (**a**–**c**): Leaf primordia; (**d**–**f**): Growing point; (**g**–**i**): Vascular strand primordia; (**j**–**l**): Bud primordia; m-o: Underground bud. The scale bars are in the lower left corner of the picture.

**Figure 5 ijms-22-08382-f005:**
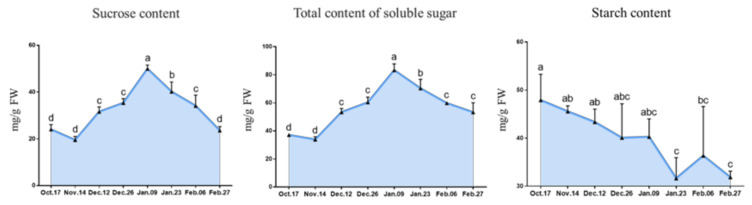
The changes in sugar and carbohydrate contents during the bud dormancy period of ‘Meiju’. Each bar represents the mean ± SEM (*n* = 3). The experimental time was from 2018 to 2019, and the data were processed by IBM SPSS Statistics 26. Different letters indicate a significant difference among different dates (*p* < 0.05).

**Figure 6 ijms-22-08382-f006:**
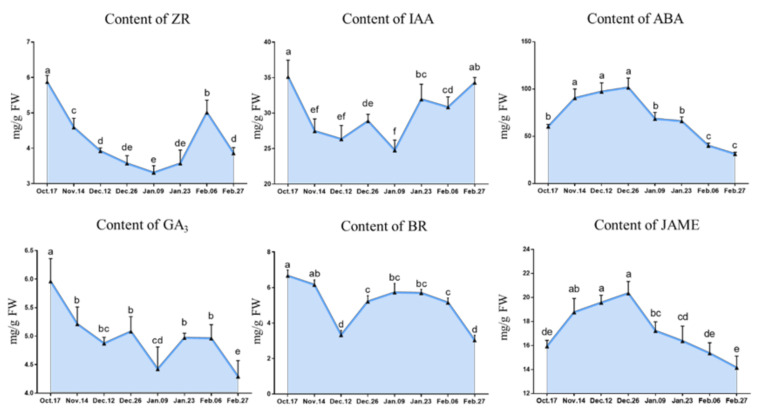
The changes in hormone contents during the underground bud dormancy period of ‘Meiju’. Each bar represents the mean ± SEM (*n* = 3). Different letters indicate a significant difference among different dates (*p* < 0.05). The experimental time was from 2018 to 2019, and the data were processed by IBM SPSS Statistics 26.

**Figure 7 ijms-22-08382-f007:**
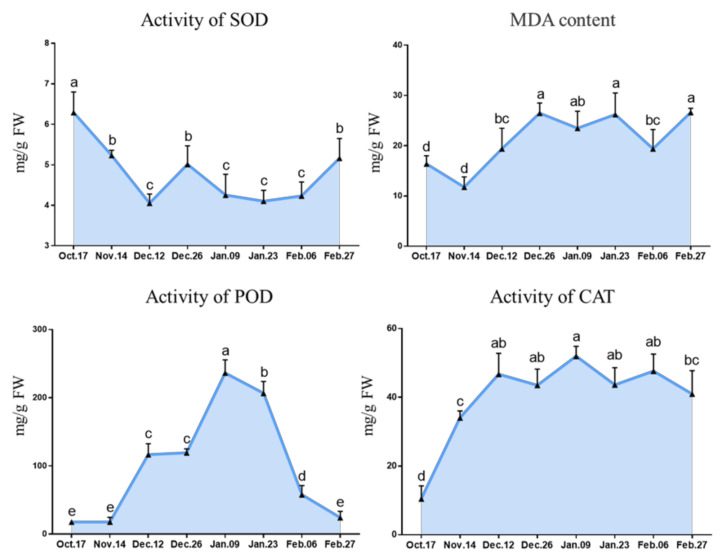
Antioxidant metabolic enzyme activity and MDA content in underground buds of ‘Meiju’ at different dormancy stage. Different letters indicate a significant difference among different dates (*p* < 0.05). The experimental time was from 2018 to 2019, and the data were processed by IBM SPSS Statistics 26.

**Figure 8 ijms-22-08382-f008:**
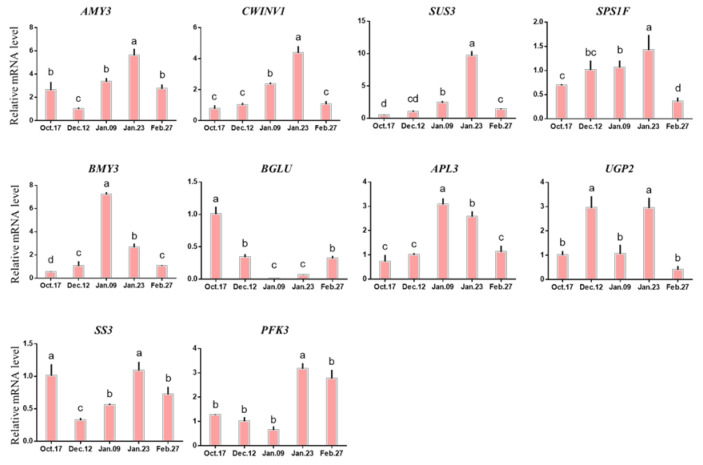
Expression of the genes involved in carbohydrate metabolism during bud dormancy of ‘Meiju’. Each bar represents the mean ± SEM (*n* = 3). Different letters indicate a significant difference among different dates (*p* < 0.05). The experimental time was from 2018 to 2019, and the data were processed by IBM SPSS Statistics 26.

**Figure 9 ijms-22-08382-f009:**
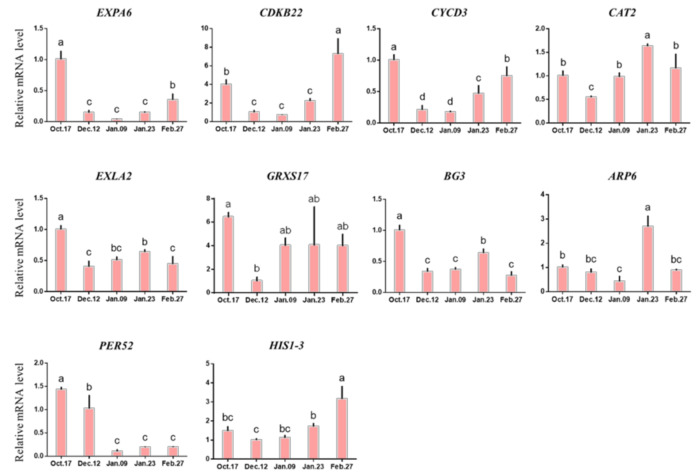
Change in the expression levels of antioxidant metabolism-related genes. Each bar represents the mean ± SEM (*n* = 3). Different letters indicate a significant difference among different dates (*p* < 0.05). The picture was drawn by GraphPad Prism 8.0 and the data were processed by IBM SPSS Statistics 26.

**Figure 10 ijms-22-08382-f010:**
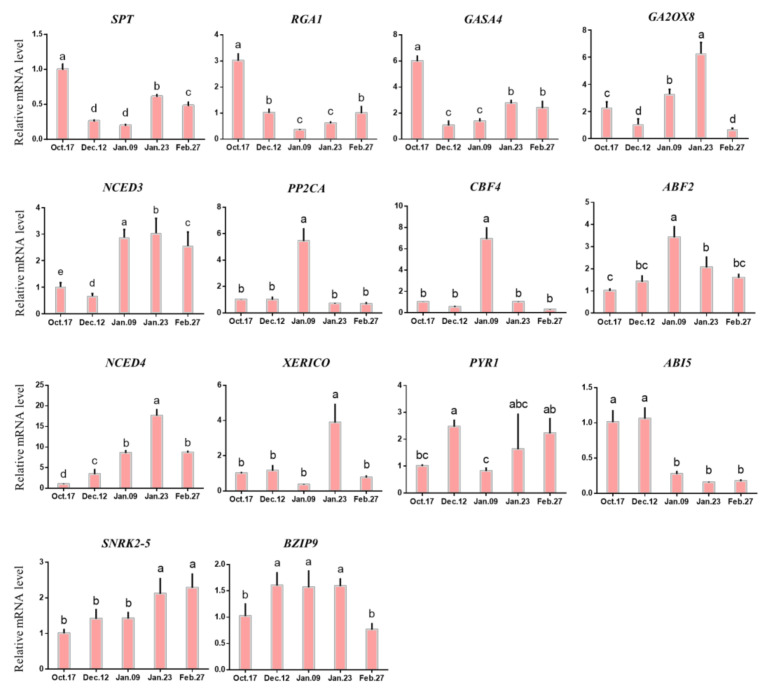
Expression of hormone metabolism-related genes in different dormancy stages. Each bar represents the mean ± SEM (*n* = 3). Different letters indicate a significant difference among different dates (*p* < 0.05). The picture was drawn by GraphPad Prism 8.0 and the data were processed by IBM SPSS Statistics 26.

**Figure 11 ijms-22-08382-f011:**
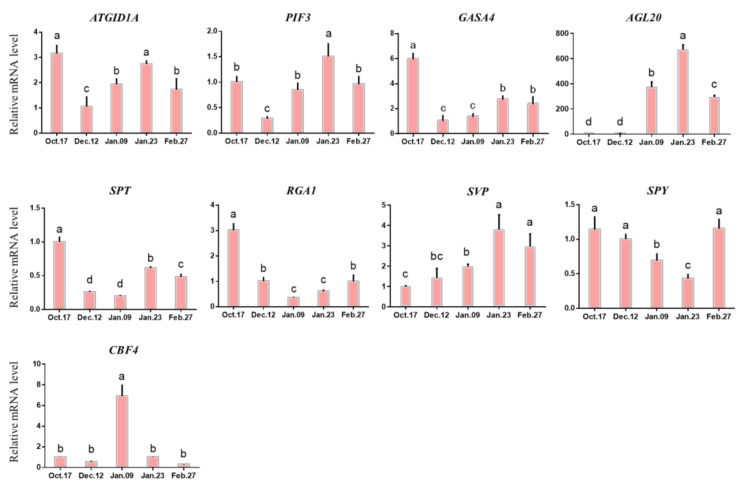
Expression of DAM-SOC1-AP1 pathway and other genes. Each bar represents the mean ± SEM (*n* = 3).Different letters indicate a significant difference among different dates (*p* < 0.05). The picture was drawn by GraphPad Prism 8.0 and the data were processed by IBM SPSS Statistics 26.

**Figure 12 ijms-22-08382-f012:**
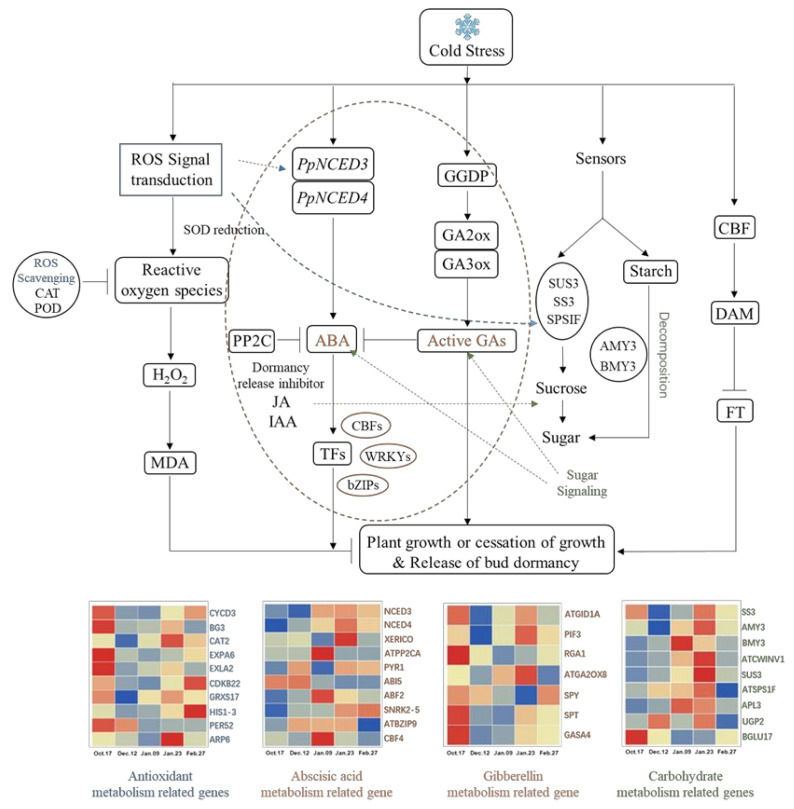
Underground bud endodormancy regulatory network. The picture was drawn by TBtools and Microsoft Office 2019. The data were processed by IBM SPSS Statistics 26. Full name and function of the genes are shown in Table 2.

**Figure 13 ijms-22-08382-f013:**
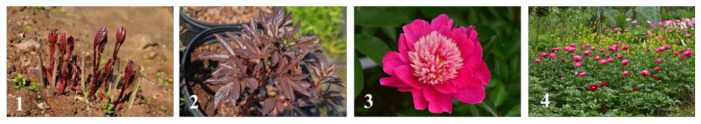
Growth of ‘Meiju’: From sprouting to flowering. ‘Meiju’ was planted outside at the Perennial Flower Resources Garden of Zhejiang University in Hangzhou (E 118°21′–120°30′, N 29°11′–30°33′).

**Table 1 ijms-22-08382-t001:** Correlation analysis of related crucial indices.

	ZR	IAA	ABA	GA_3_	BR	JAME	SUC	TSS	STA	POD	SOD	CAT	MDA	BFP	WFS	ANS	APW	APH	ADS
ZR	1.00																		
IAA	0.52	1.00																	
ABA	−0.27	−0.64	1.00																
GA_3_	0.77 *	0.35	0.26	1.00															
BR	0.41	0.02	0.16	0.66	1.00														
JAME	−0.31	−0.69	0.98 **	0.17	0.08	1.00													
SUC	−0.6	−0.48	0.07	−0.41	0.1	0.1	1.00												
TSS	−0.72 **	−0.38	−0.11	−0.63	−0.11	−0.06	0.95 **	1.00											
STA	0.52	−0.27	0.54	0.66	0.45	0.50	−0.32	−0.56	1.00										
POD	−0.73 **	−0.52	0.24	−0.4	0.08	0.23	0.94 *	0.89 **	−0.31	1.00									
SOD	0.64	0.57	−0.1	0.62	0.37	−0.15	−0.66	−0.71 *	0.5	−0.68	1.00								
CAT	−0.79 **	−0.62	0.07	−0.83 *	−0.48	0.18	0.63	0.74 *	−0.56	0.61	−0.90 **	1.00							
MDA	−0.70	0.14	−0.21	−0.58	−0.42	−0.17	0.55	0.71 *	−0.77 *	0.54	−0.36	0.49	1.00						
BPF	−0.77 *	−0.49	0.07	−0.84 *	−0.65	0.19	0.76 *	0.81 *	−0.75	0.71	−0.90 **	−0.93 **	0.74	1.00					
WFS	0.77 *	0.6	−0.13	0.87 *	0.52	−0.25	−0.57	−0.65	0.74	−0.58	0.89 **	−0.96 **	−0.51	−0.90 **	1.00				
ANS	−0.80 *	−0.49	0.14	−0.82 *	−0.6	0.26	0.7	0.76 *	−0.78 *	0.68	−0.87 *	0.93 **	0.75	0.98 **	−0.94 **	1.00			
APW	−0.88 **	−0.56	0.10	−0.90 **	−0.46	0.19	0.79 *	0.84 *	−0.80 *	0.80 *	−0.89 **	0.94 **	0.73	0.95 **	−0.95 **	0.96 **	1.00		
APH	−0.75	−0.61	0.02	−0.92 **	−0.6	0.17	0.75	0.80 *	−0.69	0.66	−0.90 **	0.98 **	0.62	0.97 **	−0.93 **	0.95 **	0.94 **	1.00	
ADS	−0.65	−0.34	−0.20	−0.80 *	−0.45	−0.06	0.75	0.81 *	−0.88 **	0.658	−0.88 **	0.90 **	0.7	0.95 **	−0.90 **	0.95 **	0.93 **	0.94 **	1.00

Note: The symbol “**” indicates extreme significance (*p* < 0.01), and the symbol “*” represents significance (*p* < 0.05). The darker the background color is, the more significant the correlation is. Blue indicates a negative correlation, and red indicates a positive correlation. The full names of the acronyms are shown in the main body of the article and in the notes of Table 2.

**Table 2 ijms-22-08382-t002:** Selection of genes related to carbohydrate metabolism, hormone metabolism and dormancy regulation.

Abbreviations	Full Names	Gene	Physiological Metabolic Pathways
*SS3*	*starch synthase 3*	*AT1G11720*	Carbohydrate transport and metabolism
*AMY3*	*α-amylase-like 3*	*AT1G69830*	Carbohydrate transport and metabolism
*BMY3*	*β-amylase 3*	*AT5G18670*	putative beta-amylase
*ATCWINV1*	*Arabidopsis thaliana* *cell wall invertase 1*	*AT3G13790*	Carbohydrate transport and metabolism
*SUS3*	*sucrose synthase 3*	*AT4G02280*	Cell all/membrane/envelope biogenesis
*ATSPS1F*	*sucrose phosphate synthase 1f*	*AT5G20280*	Cell all/membrane/envelope biogenesis
*PFK3*	*phosphofructokinase 3*	*AT4G26270*	Carbohydrate transport and metabolism
*APL3*	*/*	*AT4G39210*	Carbohydrate transport and metabolism
*UGP2*	*UDP-glucose pyrophosphorylase 2*	*AT5G17310*	Carbohydrate transport and metabolism
*BGLU17*	*β-glucosidase 17*	*AT2G44480*	Carbohydrate transport and metabolism
*NCED3*	*9-cis-epoxycarotenoid dioxygenase 3*	*AT3G14440*	Secondary metabolites biosynthesis, transport and catabolism
*NCED4*	*9-cis-epoxycarotenoid dioxygenase 4*	*AT4G19170*	Secondary metabolites biosynthesis, transport and catabolism
*XERICO*		*AT2G04240*	Involved in ABA metabolism
*ATPP2CA*	*protein phosphatase 2CA*	*AT3G11410*	Signal transduction mechanisms
*PYR1*	*pyrabactin resistance 1*	*AT4G17870*	PYR/PYL/RCAR family proteins function as abscisic acid sensors
*ABI5*	*ABA insensitive 5*	*AT2G36270*	involved in ABA signalling during seed maturation and germination
*ABF2*	*abscisic acid responsive elements-binding factor 2*	*AT1G45249*	Leucine zipper transcription factor that binds to the abscisic acid (ABA)–responsive element (ABRE) motif in the promoter region of ABA-inducible genes.
*SNRK2-5*	*SNF1-related protein kinase 2.5*	*AT5G63650*	encodes a member of SNF1-related protein kinases (SnRK2) whose activity is activated by ionic (salt) and non-ionic (mannitol) osmotic stress.
*ATBZIP9*	*Arabidopsis thaliana* *basic leucine zipper 9*	*AT5G24800*	
*CYCD3*	*cyclin d3*	*AT4G34160*	Cell cycle control, cell division, chromosome partitioning
*BG3*	*β-1,3-glucanase 3*	*AT3G57240*	encodes a member of glycosyl hydrolase family 17
*CAT2*	*catalase 2*	*AT4G35090*	Catalase domain-containing protein/Catalase-rel domain-containing protein [Cephalotus follicularis]
*EXPA6*	*Arabidopsis thaliana* *texpansin 6*	*AT2G28950*	Encodes an expansin. Involved in the formation of nematode-induced syncytia in roots of Arabidopsis thaliana.
*EXLA2*	*expansin-like a2*	*AT4G38400*	member of EXPANSIN-LIKE.
*CDKB22*	*cyclin-dependent kinase b2;2*	*AT1G20930*	Expressed in the shoot apical meristem. Involved in regulation of the G2/M transition of the mitotic cell cycle.
*GRXS17*	*Arabidopsis thaliana* *monothiol glutaredoxin 17*	*AT4G04950*	Encodes a monothiol glutaredoxin that is a critical component involved in ROS accumulation, auxin signaling, and temperature-dependent postembryonic growth in plants.
*HIS1-3*	*histone H1-3*	*AT2G18050*	encodes a structurally divergent linker histone whose gene expression is induced by dehydration and ABA. The mRNA is cell-to-cell mobile.
*PER52*	*peroxidase 52*	*AT5G05340*	Encodes a protein with sequence similarity to peroxidases that is involved in lignin biosynthesis. Loss of function mutations show abnormal development of xylem fibers and reduced levels of lignin biosynthetic enxymes.
*ARP6*	*actin-related protein 6*	*AT3G33520*	Encodes *ACTIN-RELATED PROTEIN6* (*ARP6*), a putative component of a chromatin-remodeling complex.
*GID1A*	*GA insensitive dwarf1a*	*AT3G05120*	Encodes a gibberellin (GA) receptor ortholog of the rice GA receptor gene (*OsGID1*).
*PIF3*	*phytochrome interacting factor 3*	*AT1G09530*	Transcription factor interacting with photoreceptors phyA and phyB.
*RGA1*	*repressor of GAL-3 1*	*AT2G01570*	Putative transcriptional regulator repressing the gibberellin response and integration of phytohormone signalling.
*ATGA2OX8*	*Arabidopsis thaliana* *gibberellin 2-oxidase 8*	*AT4G21200*	Encodes a protein with gibberellin 2-oxidase activity which acts specifically on C-20 gibberellins.
*SPY*	*spindly*	*AT3G11540*	Encodes a N-acetyl glucosamine transferase that may glycosylate other molecules involved in GA signaling.
*SPT*	*spatula*	*AT4G36930*	Encodes a transcription factor of the bHLH protein family. Mutants have abnormal, unfused carpels and reduced seed dormancy.
*GASA4*	*GAST1 protein homolog 4*	*AT5G15230*	Encodes gibberellin-regulated protein GASA4. Promotes GA responses and exhibits redox activity.
*CBF4*	*C-repeat-binding factor 4*	*AT5G51990*	This gene is involved in response to drought stress and abscisic acid treatment, but not to low temperature.
*SPL9*	*squamosa promoter binding protein-like 9*	*AT2G42200*	Encodes a putative transcriptional regulator that is involved in the vegetative to reproductive phase transition.
*SVP*	*short vegetative phase*	*AT2G22540*	Encodes a nuclear protein that acts as a floral repressor and that functions within the thermosensory pathway.
*AGL20*	*agamous-like 20*	*AT2G45660*	Controls flowering and is required for CO to promote flowering.
*AP1*	*apetala1*	*AT1G69120*	Floral homeotic gene encoding a MADS domain protein homologous to SRF transcription factors.
*GRF2*	*general regulatory factor 2,*	*AT1G78300*	G-box binding factor GF14 omega encoding a 14-3-3 protein The mRNA is cell-to-cell mobile.

Note: Full names and notes of genes in the table are from “TAIR—Home Page (arabidopsis.org)”. “/” indicates the information could not be found.

## Data Availability

No new data were created or analyzed in this study. Data sharing is not applicable to this article.

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
