# Peer review of "Chilling Requirement Validation and Physiological and Molecular Responses of the Bud Endodormancy Release in Paeonia lactiflora ‘Meiju’"

_ijms, 2021, doi:10.3390/ijms22168382_

Round 1
Reviewer 1 Report
The aim of the study was to determine the chilling requirement for Paeonia lactiflora ‘Meiju’ and regulation of dormancy on the base of morphological, physiological and molecular responses. The subject of the study is interesting, but in the current form presents deficiencies. My comments are as follows:
- Title – please add the name of cultivar
Introduction doesn’t show the back ground for the experiment. The regulation of dormancy as well herbaceous as tree peony has been studied for at least 30 years. The Authors should primarily show what factors influenced regulation of dormancy in peonies, what has been done, and what the performed experiments will contribute.
- Material and Methods
Line 457 – please add what low CR traits means and what is annual growth cycle of P. lactiflora ‘Meiju’.
Line 464 - there is no information what temperature was in the greenhouse. Temperature is a key factor for all metabolic, hormonal and molecular changes occurring in buds. A major shortcoming is the lack of control (outdoors).
Line 486 – please add information about the starting point of cooling capacity assessment
Line 469 – “in different growth stages” please provide what growth stages and associated analysis dates
- Results
Line 127 please add time when ‘Meiju’ defoliated.
Fig.1A- it is not the result of the study and should be included in the M&M section. Current Fig. 1B should be shown below the description of the results presented in figure.
Line 129 “new buds sprouted around Jan. 9” – outdoors? If so, it means that those transferred after that date were already active. Please explain the results in Figure 3, where the number of weeks until the first plant sprouted in the greenhouse (WFS) is presented. How many buds initiated growth 2 weeks after the potted crown were transferred to the greenhouse on Jan.9, Jan. 23 and on February?
There is lack of result of WAS (line 469). Did the Authors observe differences in flowering depending on cooling time?
The description of gene expression is poor in some place, e.g. line 242-243, line 254-255
- Discussion
Line 293 - please add what is CR for other Peonia lactiflora cultivars by the UT model
Carbohydrate metabolism- please refer to the study Mornys and Cheng (2013) Seasonal changes in endogenous hormone and sugar contents during bud dormancy in tree peony. J App.Hort 15; 159-165
Line 410-411 “the change in external temperature …..causing damage the plant ” –it is not clear what temperature? Please also refer to the temperature in the greenhouse.
Line 429- the formulation “In this study, Paeonia lactiflora was introduced from high latitudes to low latitudes…. „ seems to be far-reaching
Line 438 It is not clear what ecological dormancy means?
Line 434-450 adds nothing and I propose that it be removed. First paragraph can be moved to the Conclusions chapter
Conclusions - What are the lessons for horticulture? How many weeks and what temperature are necessary to break dormancy and to induce flowering Paeonia lactiflora ‘Meiju’?
Author Response
Dear reviewer,
I have finished revising the manuscript. Thank you for your guidance. I hope this manuscript could be accepted!
best regards

Reviewer 2 Report
The paper 'Chilling requirement validation and physiological and molecular responses of the bud endodormancy release in Paeonia lactiflora Pall.' by Runlong Zhang , Xiaobin Wang , Xiaohua Shi , Lingmei Shao , Tong Xu , Yiping Xia , Danqing Li * , Jiaping Zhang attempts to study the effects of temperature during bud dormancy. It is a comprehensive attempt targeting different pathways of bud dormancy control.
The paper is well written although there are some typos and English errors that require attention. Regarding the main body of results and discussion I have some questions and suggestions:
Line 139 - please explain the indices in the text and explain how they were affected through the experience.
Line 158 - shouldn't be endodormancy?
Line 160 - Nuclear intensity differences is a little dubious. Could you find a way to quantify?
Line 164 - what do you mean by mature cells? And which types of morphologies?
I advise that 2.1.2 should be integrated in 2.1.3 to sustain your choice of dates for tissue analysis
Figure 3 - Were there no points between 14 NOV and 12 DEC? It would improve the data analysis as there is sometimes a big gap spanning these dates.
Line 190- Why did you use OCT17?
Line 210 - Change observed for determined
Line 221 - I could not find the data that sustained the correlation analysis. All the dates were used or only Jan 9?
Genes Table - CBF4 is repeated twice
Line 238- How do you determined ecodormaacy release?
Line 242. I would advice changing the generalisation. Is there data from NOV when the chilling hours start to accumulate?
Line 249 - mention which genes
2.4.4 - I advice to speak of the genes associated to hormones than to speak about trends. It would be easier to process
The figure legends should be more complete.
Line 287 - What are typical materials?
Line 298 - What changes have you made to your previous morphological data to support the decrease in CR?
Line 320 - What is transformation in bud physiology dormancy?
Line 359 - no discussion is made on the genes expression and hormonal content of GA, IAA, JA, ZR and BR. And no discussion about the correlation between ABA and ZR.
Line 365 - eco or endodormancy?
Line 370 - paradormancy?
Lines 391/2 - Bad English, please rephrase.
Line 394 - What previous evidence suggests that ROS leads to an increase in MDA?
line 399 - How do the NCEDs expression increase and ABA decreases? Please explain.
Line 409 - which plants?
Line 411 - How is MDA content correlated with temperature?
Author Response

(The authors gave the same response as above.)

Round 2
Reviewer 1 Report
The Authors have made a lot of valuable changes. However, I have a few minor comments:
1/Key words should be in alphabetical order
2/ The title of Figure in Introduction is missing. Hence the numbering of the other figures should be changed and their citation in the text corrected
3/ Figures should be placed below the information in the text. Please do not refer to figures 2-13 in the Introduction
4/ 2.1.1 Please correct “osing “In autumn, initial periods of more than 50% leaves osing?”
5/ Discussion – the name of cultivar should be placed in single quotation marks; please correct (“Zhuguang”, “Qiaoling”, “Qihua Lushuang”, “Fen Yunu”)
6/ The Latin name of the species should be written in capital letters; please correct “arabidopsis thaliana”
Author Response
Response to Reviewer 1
We appreciate that reviewers gave us an opportunity to improve this manuscript, which named “Chilling requirement validation and physiological and molecular responses of the bud endodormancy release in Paeonia lactiflora ‘Meiju’.”
Your review and suggestions are of great importance to us. According to your request, we have revised the manuscript.
I hope that under your guidance, my manuscript will be accepted!
“1/Key words should be in alphabetical order”
Thank you for your timely correction. I have rearranged the key words in alphabetical order.
“2/ The title of Figure in Introduction is missing. Hence the numbering of the other figures should be changed and their citation in the text corrected”
Thank you for your guidance. I have arranged the pictures of the full text in order and deleted some of them according to your request.
“3/ Figures should be placed below the information in the text. Please do not refer to figures 2-13 in the Introduction”
Thank you for your suggestion. I have revised it in accordance with the requirements of “Figures should be placed below the information in the text.”
“4/ 2.1.1 Please correct “osing “In autumn, initial periods of more than 50% leaves osing?”
Thank you for your timely correction. I have changed “osing” to “losing”.
“5/ Discussion – the name of cultivar should be placed in single quotation marks; please correct (“Zhuguang”, “Qiaoling”, “Qihua Lushuang”, “Fen Yunu”)”
Thank you for your guidance. I have changed the double quotation marks of the cultivars name into single quotation marks.
“6/ The Latin name of the species should be written in capital letters; please correct “arabidopsis thaliana”
Thank you for your correction. I have changed “arabidopsis thaliana” to “Arabidopsis thaliana”.
Reviewer 2 Report
The authors addressed all my questions and made the appropriate changes to the manuscript. I advise accepting in the current form.
Author Response
Response to Reviewer 2
We appreciate that reviewers gave us an opportunity to improve this manuscript, which named “Chilling requirement validation and physiological and molecular responses of the bud endodormancy release in Paeonia lactiflora ‘Meiju’.”
I was very excited after receiving your review comments. Thank you for your guidance. Your suggestion is of great help to our promotion. I hope this manuscript will be accepted smoothly under your guidance!